# Stimulus background influences phase invariant coding by correlated neural activity

**Michael G Metzen, Maurice J Chacron***

Department of Physiology, MGill University, Montreal, Canada

**Abstract** Previously we reported that correlations between the activities of peripheral afferents mediate a phase invariant representation of natural communication stimuli that is refined across successive processing stages thereby leading to perception and behavior in the weakly electric fish *Apteronotus leptorhynchus* (Metzen et al., 2016). Here, we explore how phase invariant coding and perception of natural communication stimuli are affected by changes in the sinusoidal background over which they occur. We found that increasing background frequency led to phase locking, which decreased both detectability and phase invariant coding. Correlated afferent activity was a much better predictor of behavior as assessed from both invariance and detectability than single neuron activity. Thus, our results provide not only further evidence that correlated activity likely determines perception of natural communication signals, but also a novel explanation as to why these preferentially occur on top of low frequency as well as low-intensity sinusoidal backgrounds.

## Introduction

We have recently proposed a neural mechanism that allows for the emergence and refinement of an invariant representation of natural electrocommunication signals (i.e. chirps) occurring at different phases of a sinusoidal background beat in the weakly electric fish *Apteronotus leptorhynchus* (*Metzen et al., 2016*). This beat is caused by the interference between the quasi-sinusoidal electric organ discharges (EODs) of two fish that are in close proximity of one another and its frequency is equal to the difference between the two individual EOD frequencies. Weakly electric fish also emit communication signals called 'chirps' consisting of a transient increase in EOD frequency that occur on top of the background beat. Behavioral data shows that chirps can occur at any phase of the beat with uniform probability (*Walz et al., 2013*; *Aumentado-Armstrong et al., 2015*). Thus, a chirp with given duration and frequency increase can give rise to very different patterns of stimulation depending on the beat phase at which it occurs (*Benda et al., 2005*; *Marsat et al., 2009*; *Walz et al., 2013*, *2014*; *Metzen et al., 2016*).

We previously found that the single neuron spiking activity of peripheral electrosensory afferents did not provide a phase invariant representation as these faithfully encoded the different stimulus waveforms resulting from the same chirp occurring at different phases of the beat. However, when instead considering population activity, we found that correlations between afferent spike trains systematically increased in response to the chirp irrespective of beat phase, thereby providing a phase invariant representation. Recordings from pyramidal cells within the electrosensory lateral line lobe (ELL) receiving synaptic input from multiple peripheral afferents showed that their responses were 'locally' phase invariant (i.e. they gave similar responses only to stimulus waveforms resulting from a given chirp occurring at some but not all beat phases). Recordings from neurons within the midbrain Torus semicircularis (TS) revealed the emergence of a more 'globally' phase invariant representation as these responded to the different stimulus waveforms resulting from the same chirp occurring at

*For correspondence: maurice.chacron@mcgill.ca

**Competing interests:** The authors declare that no competing interests exist.

different beat phases through similar increases in firing rate. Finally, we showed that weakly electric fish gave similar behavioral responses to patterns of stimulation resulting from the same chirp occurring at different beat phases, suggesting that the globally phase invariant representation of chirps in midbrain is decoded and further refined downstream (*Metzen et al., 2016*).

It is important to note that our previous results were obtained using only a low background beat frequency that was not varied. However, behavioral data shows that chirps are emitted on top of beats with frequencies spanning a wide range (*Zakon et al., 2002*), which gives rise to further heterogeneities in the resulting stimulus waveforms (*Zupanc and Maler, 1993*; *Hupé et al., 2008*). Here, we investigated the effects of varying the background beat frequency on phase invariant coding and perception of chirp stimuli.

## Results

We recorded from peripheral receptor afferents and measured behavioral responses at the organismal level to stimulus waveforms resulting from a chirp with given frequency excursion and duration occurring at different phases within a background with different frequencies. In the absence of communication signals (i.e. when both EOD frequencies are constant), interference between the emitter and receiver fish's EODs gives rise to a sinusoidal amplitude modulation (i.e. the beat) (*Figure 1A, B*). Natural communication signals (i.e. chirps) occur during social encounters in which the emitter fish sends the signal to the receiver fish (*Figure 1C*). As mentioned above, the chirp consists of one fish increasing its EOD frequency for a short duration (*Figure 1D*, top red/green trace), which, when considering the resulting stimulus sensed by the receiver fish, consists of a transient perturbation of the underlying beat (*Figure 1D*, black bottom trace). Because a chirp with a given EOD frequency time course (i.e. duration and frequency excursion) can occur with uniform probability at all phases of the beat (*Walz et al., 2013*; *Aumentado-Armstrong et al., 2015*) as well as different beat frequencies (*Zakon et al., 2002*), the resulting stimulus waveforms display significant heterogeneities (*Figure 1E*) (*Metzen et al., 2016*). It is important to note that chirp stimulus waveforms are more similar to the background itself for higher beat frequencies (*Figure 1E*, compare columns). This is because the 'frequency contrast' between the beat and the chirp is then lower, which makes differentiating chirp signals from the background beat more difficult (*Figure 1F*). Despite this decrease in detectability, it is important to notice that the stimulus waveforms resulting from the same chirp occurring at different beat phases remain equally different from one another irrespective of background beat frequency, as quantified by the average distance between all possible combinations of chirp signals at a given beat frequency (*Figure 1G*).

### Single afferent activity is not phase invariant and is a poor detector of chirp occurrence for all beat frequencies

We recorded from single peripheral receptor afferents (N = 116) (*Figure 2A*) in response to different patterns of sensory input resulting from the same chirp stimulus occurring at different phases of a background beat with varying frequency. Single afferent responses consisted of patterns of increases and decreases in firing activity that strongly depended on the stimulus waveform that was presented (*Figure 2B*, blue curves; *Figure 2—figure supplement 1A*, blue raster plots). For all beat frequencies, single neuron activity strongly varied depending on where the chirp occurred within the beat cycle (*Figure 2C*, top, blue curves). Interestingly, we found that the duration of the single neuron response to chirps decreased as a function of increasing background beat frequency (*Figure 2—figure supplement 1B*, blue) despite the fact that the actual chirp duration (i.e. the time interval during which there is an increase in EOD frequency, see *Figure 1D*) is constant by construction. We therefore used time windows whose duration matched those observed experimentally for single afferent responses to quantify invariance using the same measure that was used previously (see Materials and methods) to ensure that invariance was computed using only the response to the chirp stimulus itself. We found low values for all background beat frequencies (*Figure 2D*, blue curve). This indicates that the distance between responses was always more or less equal to the distance between the corresponding stimulus waveforms.

It is important to note that single afferents displayed phase locking (*Keener et al., 1981*; *Trussell, 1999*) to the beat for higher background beat frequencies as action potentials reliably occurred only during a restricted range of phases during the beat cycle (*Figure 2B* and *Figure 2—figure*

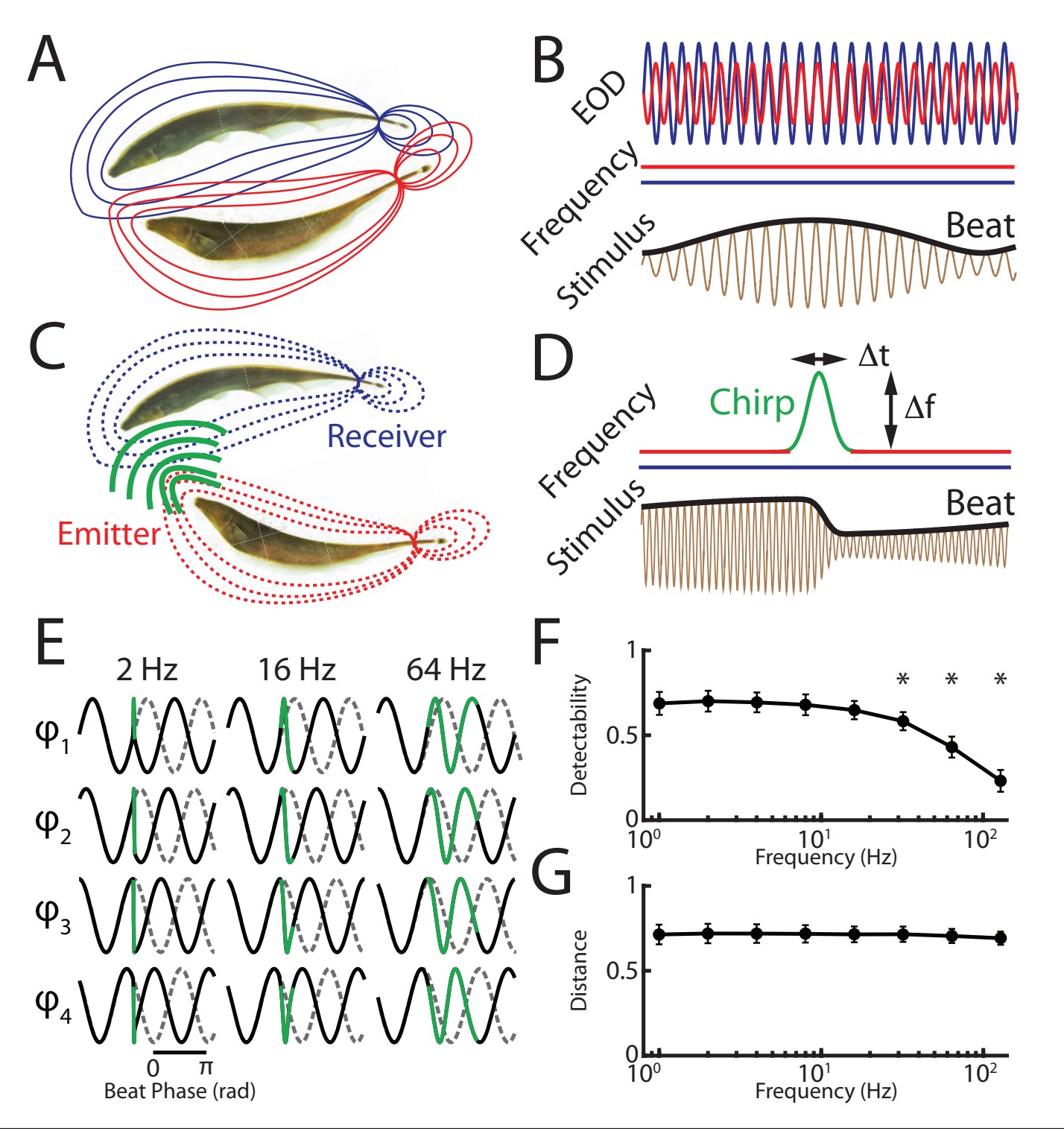

**Figure 1.** A chirp with given duration and frequency excursion gives rise to heterogeneous waveforms depending on time of occurrence within the beat cycle as well as beat frequency. (A) Two weakly electric fish with their electric organ discharges (EODs) in red and blue. (B) The EOD waveforms of both fish (top red and blue traces) show alternating regions of constructive and destructive interference when the instantaneous EOD frequencies do not vary in time (middle red and blue traces). Interference between the EODs leads to a sinusoidal amplitude modulation (i.e. a beat, bottom black trace) of the summed signal (bottom brown trace). (C) Schematic showing communication between the emitter (red) and receiver (blue) fish. (D) During a communication call, the emitter fish's EOD frequency (top red trace) transiently increased by a maximum of $\Delta f$ for a duration $\Delta t$ (top green trace) while the receiver fish's EOD frequency (top blue trace) remains constant. The communication call results in a phase reset of the beat (bottom black trace). (E)

*Figure 1 continued on next page*

*Figure 1 continued*

Ongoing unperturbed beat (gray dashed traces) and stimulus waveforms (black traces) resulting when a chirp (green) with the same frequency excursion $\Delta f$ and duration $\Delta t$ occurs at different phases during beat cycles with frequencies 2 Hz (left), 16 Hz (middle), and 64 Hz (right). We note that the waveforms of the remaining four phases are mirror images of the waveforms shown. (F) Detectability of chirps occurring at different phases of the beat cycle as a function of beat frequency. Chirps occurring at higher beat frequencies are harder to detect as they are more similar to the beat. (G) Distance between chirp waveforms occurring at different phases of the beat cycle across beat frequencies. Inter-chirp distances remain constant across the beat frequencies used. '*' indicates statistical significance to all values obtained for lower frequencies at the p=0.05 level using a one-way ANOVA with Bonferroni correction.

The following source data is available for figure 1:

**Source data 1.** Source data for *Figure 1*.

*supplement 2A,B*). Such phase locking results from the fact that afferents display high-pass filtering properties (*Bastian, 1981*; *Xu et al., 1996*; *Chacron et al., 2005*). As such, background beats with higher frequencies will elicit greater modulations in firing rate around the baseline (i.e. in the absence of stimulation) value, thereby increasing the probability of rectification (i.e. cessation of firing). Indeed, the range of stimulus phases during which the firing rate is null increases with beat frequency (*Figure 2B*, compare blue traces). It is these periods during which there is complete cessation of firing that decrease the distance between responses to chirps occurring at different beat phases, thereby slightly increasing the invariance score for higher (>32 Hz) beat frequencies (*Figure 2D*, blue).

When instead considering detectability, we found that single afferent responses to the chirp are as variable as the responses to the beat itself for all beat frequencies (*Figure 2E*, blue). This is not surprising given the results above showing that the relative distance between chirp stimulus waveforms remain constant across beat frequencies. However, the modulations in firing rate caused by the chirp and the background beat were similar irrespective of beat frequency. Consequently, detectability of chirps as quantified from single afferent activity was low for all beat frequencies (*Figure 2E*, blue).

## Correlated afferent activity is only phase invariant and a good detector of chirp occurrence for low beat frequencies

We next investigated the effects of varying background beat frequency on afferent responses at the population level. To do so, we quantified pairwise correlations between afferent activities as a function of time during the beat and the chirp as done previously (see Materials and methods) (*Metzen et al., 2016*). We found that the duration of the response to a chirp using correlated activity decreased as a function of increasing background beat frequency (*Figure 2—figure supplement 1A,B*, purple) and therefore used time windows whose duration matched those observed experimentally for correlated activity to quantify invariance using the same measure as done previously (see Materials and methods).

For low (<16 Hz) background beat frequencies, we found that afferent spiking activity was synchronized either through large increases in firing rate or cessation of firing after chirp onset but not during the beat (*Figure 2—figure supplement 1A*). Thus, there was an increase in correlation after chirp onset irrespective of the beat phase the chirp occurred at for low background beat frequencies (*Figure 2B,C*, *Figure 2—figure supplement 1A*, purple traces), consistent with our previous results. For higher background beat frequencies (>32 Hz), afferent activity was still synchronized either through large increases in firing rate or cessation of firing after chirp onset. However, afferents also displayed strong phase locking as mentioned above, thereby causing increased synchrony through cessation of firing during the beat itself at the population level (*Figure 2—figure supplement 2A, B*). Thus, afferent activities became more correlated with one another only for some portions of the beat cycle (i.e. those for which there is cessation of firing) as the beat frequency is increased (*Figure 2B*, purple traces). This phase dependency had two effects. First, we found that correlated activity no longer provided a phase invariant representation of chirp stimuli for higher beat frequencies (*Figure 2C,D*, purple). Indeed, correlations between responses to the chirp then became more alike to those found during the beat, which is expected as the corresponding stimulus waveforms

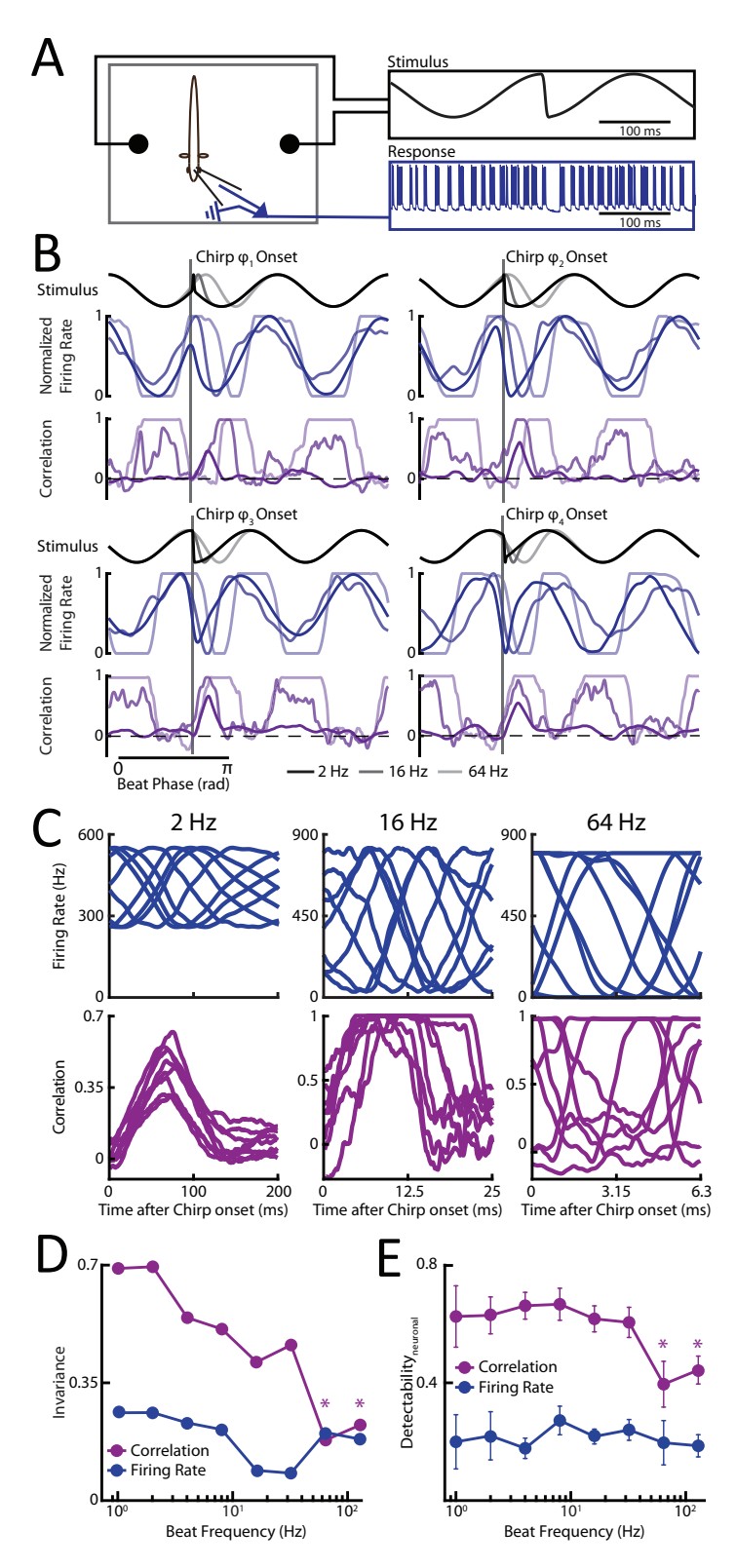

**Figure 2.** Phase invariant coding of chirps by correlated but not single neuron activity is best at low and deteriorates for higher beat frequencies. (**A**) Schematic showing the experimental setup. (**B**) Example stimulus waveforms (top) for chirps occurring at different beat frequencies (black: 2 Hz; dark gray: 16 Hz; light gray: 64 Hz), normalized firing rates (middle, blue), and spike count correlations (purple) from example afferent pairs. (**C**). Population-averaged firing rate responses (top, blue) and spike count correlation as a function of time (bottom, purple) to the different stimulus waveforms

*Figure 2 continued on next page*

*Figure 2 continued*

occurring at a beat frequencies of 2 Hz (left), 16 Hz (middle), and 64 Hz (right). Zero indicates the time at chirp onset. (D) Invariance score computed from single afferent activity (blue) and from correlated activity (purple) for all phases as a function of background beat frequency. (E) Detectability of chirp waveforms over the beat for population-averaged firing rate responses (blue) and spike count correlation (purple) as a function of background beat frequency. '*" indicates statistical significance from all values obtained for frequencies <64 Hz at the p=0.05 level using a one-way ANOVA with Bonferroni correction. We note that, in this and other figures, all neural responses are shifted to the left by 9 ms in order to account for known axonal transmission delays.

The following source data and figure supplements are available for figure 2:

**Source data 1.** Source data for *Figure 2*.
**Figure supplement 1.** Responses to small chirps occurring on top of beats with varying frequencies.
**Figure supplement 1—source data 1.** Source data for *Figure 2—figure supplement 1*.
**Figure supplement 2.** Phase locking in primary afferents to different background beat frequencies.
**Figure supplement 2—source data 1.** Source data for *Figure 2—figure supplement 2*.

are more similar to one another. The before mentioned phase dependency of correlated activity during the beat is then 'inherited' when considering responses to chirps, causing dissimilarity (*Figure 2C*, purple). Consequently, we found that invariance values strongly decreased as background beat frequency increased (*Figure 2D*, purple). We therefor conclude that phase locking in single afferent responses is strongly detrimental to phase invariant coding by correlated activity.

Second, the phase dependency of correlated activity during the background beat also had a strong effect on response detectability to chirps (*Figure 2E*, purple). Specifically, correlated activity during the beat became more similar to those elicited by the chirp as background beat frequency is increased (*Figure 2B*, purple), thereby decreasing detectability by correlated activity (*Figure 2E*, purple).

## Correlated afferent activity is a good predictor of behavior when considering both phase invariance and detectability for all beat frequencies

So far, we have shown that the increased phase locking in afferents observed for higher background beat frequencies has strong detrimental effects on detectability and phase invariant coding of chirps by correlated afferent activity. Thus, if correlated afferent activity is indeed used by downstream neurons in order to give rise to perception, then we should find that behavioral responses to chirps for higher background beat frequencies should: (1) no longer be invariant and; (2) be less likely to occur reflecting lower detectability.

To test these hypotheses, we recorded behavioral responses to patterns of stimulation resulting from the same chirp occurring at different phases within a beat cycle with different frequencies (N = 33 fish; n = 3467 chirps) (*Figure 3A*) using a previously established behavioral paradigm that measures echo chirp responses (*Zupanc et al., 2006*; *Hupé et al., 2008*; *Metzen et al., 2016*) (see Materials and methods). We found that the characteristics (i.e. frequency excursion and duration) of the chirps elicited for different background beat frequencies were not significantly different from one another (*Figure 3—figure supplement 1*). Confirming hypothesis 1), we found that echo responses were more dissimilar to one another for high beat frequencies (*Figure 3B*). Consequently, there was a strong decrease in behavioral invariance at background beat frequencies >32 Hz (*Figure 3C*, brown). Further evidence strongly suggesting that correlated afferent activity is decoded by downstream neurons comes from a good match between invariance scores obtained from correlated activity and behavior (*Figure 3C*, compare brown and purple). In contrast, there was a poor match between invariance as computed by single afferent activity and behavior (*Figure 3C*, compare brown and blue). In order to quantify detectability, we computed the echo response rate as a function of background beat frequency. We found that there was a strong decrease in echo response

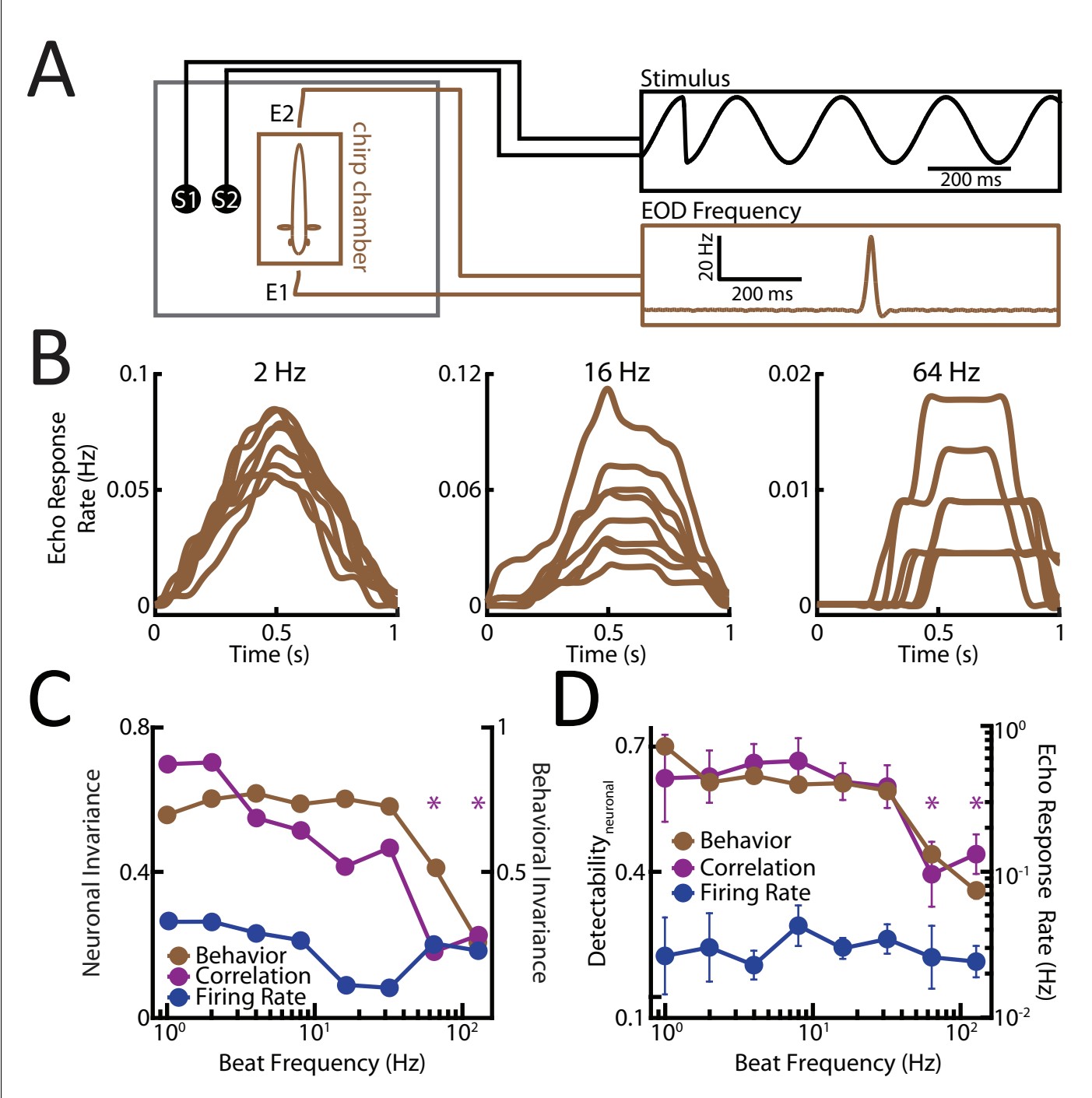

**Figure 3.** Detectability and invariance of perception are best predicted by correlated afferent activity for different beat frequencies. (**A**) Experimental setup. Each fish (N = 33) was placed in an enclosure within a tank (chirp chamber). Stimuli were applied via two electrodes (S1 and S2) perpendicular to the fish's rostro-caudal axis. The fish's EOD frequency was recorded by a pair of electrodes positioned at the head and tail of the animal (E1 and E2). Behavioral responses consisted of communication stimuli characterized by transient increases in EOD frequency in response to the presented stimulus. (**B**) Population-averaged time-dependent behavioral response rates in response to chirps occurring at different phases of the beat cycle for beat frequencies of 2 Hz (left), 16 Hz (middle), and 64 Hz (right). (**C**) Population-averaged behavioral invariance scores computed from behavioral responses obtained for different beat frequencies (brown) in comparison to the neuronal invariance scores using correlated activity (purple) and single units (blue). Note that the behavioral invariance (brown) across beat frequencies follows the one obtained for correlated activity (purple) but not single units (blue). (**D**) Chirp rate as a measure of echo response to the chirp waveforms played (brown) compared to chirp detectability computed for correlated activity (purple) and single unit firing rate (blue) as a function of background beat frequency. Note that the behavior (brown) matches the correlated activity

*Figure 3 continued on next page*

*Figure 3 continued*

(purple). '*' indicates statistical significance to all values obtained for frequencies <64 Hz at the p=0.05 level using a one-way ANOVA with Bonferroni correction.

The following source data and figure supplements are available for figure 3:

**Source data 1.** Source data for *Figure 3*.

**Figure supplement 1.** Characteristics of real chirps do not vary across baseline beat frequencies.

**Figure supplement 1—source data 1.** Source data for *Figure 3—figure supplement 1*.

rate as a function of increasing background beat frequency (*Figure 3D*, brown), thereby confirming hypothesis 2). Importantly, there was a strong match between echo response rate and detectability as computed from correlated but not single afferent activity (*Figure 3D*, compare brown, purple and blue).

## Discussion

### Summary of results

We investigated how changes in background beat frequency affected detectability and phase invariant coding of small chirps. As the signal-background frequency contrast decreased by increasing background beat frequency, we found that responses to the signal became more similar to those of the background, thereby decreasing signal detectability at the population level of peripheral afferents. We also found that single afferents tended to display phase locking as the background beat frequency increased due to their high-pass tuning properties. Such phase locking caused afferent responses to be more identical to one another during some portions of the background beat cycle, thereby causing correlations to vary widely throughout. As a consequence of phase locking and that responses to the chirp signal are more similar to those to the beat background, increasing background beat frequency decreased phase invariant coding by correlated afferent activity. These changes in neural responses were accompanied by similar changes in behavioral responses at the organismal level. Indeed, not only did detectability decrease, but behavioral responses (i.e. perception) were also no longer phase invariant for high background beat frequencies. Overall, we found that correlated afferent activity was a good predictor of behavioral performance, thereby providing further evidence supporting the hypothesis that correlated activity is decoded by higher brain centers in order to give rise to behavior. However, we cannot exclude the possibility that the changes in behavior were due to another computation being performed in the electrosensory brain. Further studies recording from higher brain areas in response to small chirps occurring on top of higher beat frequencies are needed to investigate whether there is a causal link between correlated peripheral afferent activity and behavioral responses.

### Effects of signal background on coding of small chirps

Our results provide novel explanations for previous observations. Specifically, it is well known that the small chirp stimuli considered here occur preferentially on top of low frequency beat backgrounds (*Bastian et al., 2001*; *Engler and Zupanc, 2001*; *Zakon et al., 2002*). Although it is clear that the frequency contrast between the background beat and small chirps decreases as the background beat frequency increases, previous studies have mostly considered the consequences on single neuron activity both at the periphery and more centrally (*Benda et al., 2005*; *Marsat et al., 2009*; *Walz et al., 2014*). Here, we have shown that it is not single peripheral afferent activity that can accurately predict behavioral responses but rather correlations between the spiking activities of multiple afferents. We have also shown that the decreased signal detectability was due to the greater tendency of peripheral afferents to display phase locking to higher background beat frequencies, which is a direct consequence of their high-pass frequency tuning characteristics (*Bastian, 1981*; *Xu et al., 1996*; *Chacron et al., 2005*). Such phase locking causes increased correlations

among afferents during some portions of the background beat for which both afferents do not fire action potentials. This greater variability in correlation throughout the beat cycle then decreases signal detectability. Thus, our results support the hypothesis that small chirps occur more rarely on top of beats with higher frequencies because such chirps cannot be encoded and perceived irrespective of the beat phase at which they occur. Alternatively, it is also possible that mechanisms enabling phase invariant coding and perception of small chirps occurring on top of high frequency beats did not evolve because such stimuli occur relatively infrequently in the first place. Further studies are needed to test which hypothesis is correct.

Previous studies have also shown that small chirps are preferentially elicited when both fish are located farther away from one another (*Hupé and Lewis, 2008*). Indeed, such stimuli seldom occur when both fish are in close vicinity (e.g. biting distance). Our results also provide an explanation for this behavioral observation. In this context, it is important to realize that the background beat contrast (i.e., how strong the beat amplitude is relative to each animal's unmodulated electric field amplitude) increases as distance between both fish decreases (*Yu et al., 2012*; *Fotowat et al., 2013*; *Metzen and Chacron, 2014*) and there is a lot of interest in understanding how the time-varying beat amplitude is encoded by electrosensory neurons (*Savard et al., 2011*; *McGillivray et al., 2012*; *Metzen and Chacron, 2015*; *Metzen et al., 2015*; *Huang and Chacron, 2016*; *Huang et al., 2016*; *Martinez et al., 2016*; *Zhang and Chacron, 2016*). In particular, previous studies have found that peripheral afferents will display phase locking in response to low beat frequencies when the beat amplitude is high enough, such as when both animals are right next to one another but not when located further apart (*Nelson et al., 1997*; *Kreiman et al., 2000*; *Metzen and Chacron, 2014, 2015*). Thus, while the beat amplitudes used in this study are in agreement with those experienced when both fish are located further apart (*Metzen and Chacron, 2014*) such that peripheral afferents did not display significant phase locking, it is clear that these same afferents will do so at low-frequency beat backgrounds at higher amplitudes. We predict that such phase locking will be detrimental not only on signal detectability but also on phase invariant coding for the same reasons that we found for higher background beat frequencies. If such predictions are correct, then they would provide an explanation as to why small chirps are preferentially emitted when the animals are located some distance away from one another. Further studies are needed to verify this interesting prediction and are beyond the scope of the current advance that focused on varying the background beat frequency.

## Implications for other systems

It is likely that our results will have implications for other systems and species. This is because the segregation of behaviorally relevant signals from a varying background is an important task that many sensory systems must solve (auditory system: [*Brumm and Todt, 2002*; *Lohr et al., 2003*; *Woolley et al., 2005*; *Ronacher et al., 2008*; *Berti, 2013*]; visual system: [*Born et al., 2000*; *Olveczky et al., 2003*]; electrosensory system: [*Zakon et al., 2002*; *Vonderschen and Chacron, 2009*]). Moreover, correlations between the activities of neighboring neurons are observed ubiquitously in the brain (*Averbeck et al., 2006*) and are dynamically regulated by several factors such as attention (*Cohen and Maunsell, 2009*), behavioral state (*Vaadia et al., 1995*) and stimulus statistics (*Chacron and Bastian, 2008*; *Litwin-Kumar et al., 2012*; *Ponce-Alvarez et al., 2013*; *Simmonds and Chacron, 2015*). Moreover, it is well known that neurons will display phase locking in response to sinusoidal stimulation (*Johnson, 1980*), thereby enhancing their response selectivity (*Lee et al., 2005*; *Schroeder and Lakatos, 2009*). Finally, we note that feature invariance has been demonstrated across sensory systems (visual: [*Zoccolan et al., 2007*]; auditory: [*Bendor and Wang, 2005*]; olfactory: [*Martelli et al., 2013*]). Further studies are needed to investigate the role of correlated activity and phase locking toward explaining the invariance of neural and perceptual responses to a given behaviorally relevant stimulus feature occurring in different contexts for these other sensory modalities (e.g. hearing someone's voice on top of different background noises).

## Materials and methods

### Ethics statement

All experimental procedures were approved by McGill University's animal care committee under protocol number 5285.

### Animals

*Apteronotus leptorhynchus* specimens were acquired from tropical fish suppliers and acclimated to laboratory conditions according to published guidelines (*Hitschfeld et al., 2009*).

### Surgery and recordings

Surgical procedures have been described in detail previously (*Toporikova and Chacron, 2009*; *Vonderschen and Chacron, 2011*; *McGillivray et al., 2012*; *Deemyad et al., 2013*; *Metzen et al., 2016*). Briefly, animals (N = 11) were injected with tubocurarine chloride hydrate (0.1–0.5 mg) before being transferred to an experimental tank and respirated with a constant flow of water over their gills (~10 ml/min). A small craniotomy (~5 mm$^2$) was made above the hindbrain after application of local anesthetic (5% lidocaine gel). We used 3M KCl-filled glass micropipettes (30 MΩ resistance) to record from electroreceptor afferent axons as they enter the ELL (N = 116) (*Savard et al., 2011*; *Metzen and Chacron, 2015*; *Metzen et al., 2015*, *2016*). Recordings were digitized at 10 kHz (CED Power 1401 and Spike 2 software, Cambridge Electronic Design) and stored on a computer for subsequent analysis.

### Stimulation

Stimulation was similar to that used in our previous study (*Metzen et al., 2016*). We here focused on a subtype of electrocommunication signals, namely type II or small chirps, as they occur most frequently for *Apteronotus leptorhynchus* (*Hagedorn and Heiligenberg, 1985*; *Hupé et al., 2008*; *Triefenbach and Zakon, 2008*). Small chirps are characterized as a short duration (10–20 ms) increase in EOD frequency of about 60–150 Hz and can thus be considered 'high-frequency transients' (*Zupanc and Maler, 1993*; *Engler and Zupanc, 2001*); see also *Walz et al. (2013)* for review). We generated chirps (frequency increase: 60 Hz, duration: 14 ms; see *Metzen et al. (2016)* for details) on top of a sinusoidal beat and systematically varied both the beat phase at which the chirp occurred between 0 and 315 deg in increments of 45 deg as well as the beat frequency between 1 and 128 Hz. Specifically, we used $f_{beat}$ = 1, 2, 4, 8, 16, 32, 64 and 128 Hz. The same stimuli were used to elicit both neural and behavioral responses. The stimulus intensity was set such that electrosensory afferents did not display non-linear responses such as rectification for beat frequencies up to 10 Hz and ranged between 0.14 mV/cm and 0.61 mV/cm (average: 0.34 ± 0.15 mV/cm). Importantly, the stimulus intensity was held constant for all stimuli used to elicit responses from a given afferent. The stimulus intensities used for behavioral experiments were similar to those used for recording from afferents.

### Analysis

All analyses were performed using custom-built routines in Matlab (The Mathworks, Natick, MA), these routines are freely available online.

### Electrophysiology

Action potential times were defined as the times at which the signal crossed a suitably chosen threshold value. From the spike time sequence, we created a binary sequence $R(t)$ with binwidth $\Delta t$ = 0.5 ms and set the content of each bin to be equal to the number of spikes which fell within that bin. We quantified the non-linear responses of each afferent to the beat frequency by assessing a phase locking index that determines the probability of rectification by calculating the ratio of bins of the corresponding phase histogram (*Massot et al., 2012*; *McGillivray et al., 2012*) that fall below a given threshold compared to the overall bins of the phase histogram. The threshold was set to be 0.05% of the mean spike count of a given phase histogram. A low value indicates a low amount of phase locking (i.e. spikes are distributed across all beat phases), whereas a high value indicates that spikes occur at a preferred phase within the beat cycle (i.e. a value of one means that all spikes occur

at the same phase). We note that other measures such as vector strength (*Mardia and Jupp, 1999*) can be nonzero even when the sinusoidal stimulus elicits sinusoidal modulations in firing rate around the baseline value: a situation that does not elicit phase locking according to our definition because spiking occurs throughout the stimulus cycle as there is then a linear relationship between the stimulus and the firing rate response. PSTHs were obtained by averaging the neural responses across repeated presentations of a given stimulus with binwidth 0.1 ms and were smoothed with a 6 ms long boxcar filter as done previously (*Metzen et al., 2016*).

## Correlation between the spiking activities of electrosensory afferents

We computed the spike count correlation between the spiking responses $R_i(t)$ and $R_j(t)$ of neurons $i$ and $j$ as done in previous studies (*de la Rocha et al., 2007*; *Litwin-Kumar et al., 2012*). For this, we used a sliding window (sliding time = $0.001/f_{beat}$) of length $T$ that was set to be equal to 12.5% of the corresponding beat period (1 Hz: 125 ms; 2 Hz: 62.5 ms; 4 Hz: 31.25 ms; 8 Hz: 15.63 ms; 16 Hz: 7.81 ms; 32 Hz: 3.91 ms; 64 Hz: 1.95 ms; 128 Hz: 0.98 ms). This was done to properly estimate responses to the chirp whose duration decreases with increasing beat frequency (*Figure 2—figure supplement 1A,B*). To compute the spike count, $T$ was further discretized into five bins. From the series of spike counts, the spike count correlation was calculated according the following equation:

$$\rho(n_i, n_j) = \frac{Cov(n_i, n_j)}{\sqrt{Var(n_i)Var(n_j)}} \tag{1}$$

where $n_i$, $n_j$ are the spike count series obtained for neurons $i$ and $j$, respectively, $Cov(\ldots)$ is the covariance, and $Var(\ldots)$ is the variance (*Perkel et al., 1967*; *de la Rocha et al., 2007*; *Litwin-Kumar et al., 2012*; *Doiron et al., 2016*). We note that we obtained similar results when using other methodology as done in previous studies (*Metzen et al., 2015*, *2016*).

## Quantifying neural response invariance

The phase invariance score to the different chirp waveforms across all beat frequencies was computed similarly as described in detail in our previous study (*Metzen et al., 2016*) using a distance metric (*Aumentado-Armstrong et al., 2015*). Briefly, invariance scores were computed from the population averaged responses of individual cells for a given chirp waveform within a time window whose duration was computed from a power-law fit through the response duration of our primary afferent population to chirps at different beat frequencies (*Figure 2—figure supplement 1B*, blue). Qualitatively similar invariance scores were obtained when using window sizes obtained from the response duration of correlated activity (*Figure 2—figure supplement 1C*, compare filled and hollow circles). We computed phase invariance for correlated activity as described for the single neuron responses except that we used the time course of the varying spike count correlation as an input and a time window whose size was computed from a power-law fit through the response duration of the correlated activity to chirps at different beat frequencies (*Figure 2—figure supplement 1B*, purple). This was done to ensure that only the response to the chirp was accounted for in the calculation, as including the response to the beat stimulus itself decreases invariance (see *Figure 2—figure supplement 1B,C* of *Metzen et al., 2016*). We note that changing the size of the analysis window based on beat frequency is not physiologically unrealistic, as previous studies have shown that the integration time window of sensory neurons can change based on stimulus attributes (*Reid et al., 1992*; *Bair and Movshon, 2004*; *Prescott and De Koninck, 2005*; *Assisi et al., 2007*; *Butts et al., 2007*; *Ratté et al., 2013*). Moreover, it is also possible that information carried by afferent populations is decoded by downstream neurons with different integration time windows.

## Distance

To determine the distance between different waveforms, we used a metric as done previously (*Aumentado-Armstrong et al., 2015*; *Metzen et al., 2016*):

$$D(x,y) = \frac{\sqrt{\left\langle [x - \langle x \rangle - y + \langle y \rangle]^2 \right\rangle}}{max\left[\frac{max(x)-min(x)}{\sqrt{2}}, \frac{max(y)-min(y)}{\sqrt{2}}\right]} \quad (2)$$

where <...> denotes the average over a time window equal 14 ms (i.e. the chirp duration) after chirp onset.

### Detectability

To determine the detectability of a chirp with a specific identity within the ongoing beat, we computed the distance $D(x,y)$ between the chirp waveform and the corresponding beat waveform when no chirp would have occurred over the duration of a full cycle of the respective beat frequency. A value of one indicates perfect detectability, whereas a value of zero indicates that the chirp waveform is identical to the beat waveform.

The neuronal detectability of a chirp (using either single unit firing rate or correlated activity) was computed using:

$$Detectability_{neuronal} = abs\left(\frac{R_{chirp} - R_{beat}}{R_{chirp} + R_{beat}}\right) \quad (3)$$

where $R_{chirp} = R_{maximum} - R_{minimum}$ (i.e. the difference between the maximum and minimum values of the response) in a window, whose size was taken based on the duration of the experimentally observed responses either using correlation or single unit firing rate (see *Figure 2—figure supplement 1B*), respectively, and $R_{beat} = R_{maximum} - R_{minimum}$ (i.e. the difference between the maximum and minimum values of the response to the undisturbed beat) during one beat cycle, respectively. Qualitatively similar results for chirp detectability using single unit firing rate were obtained when taking window sizes obtained from the response duration of correlated activity (*Figure 2—figure supplement 1D*, compare filled and hollow circles).

### Behavior

Echo responses to chirps occurring at each beat frequency were measured in the same way as described previously (*Metzen et al., 2016*). Briefly, each fish (N = 33) was restrained in a 'chirp chamber' as described previously (*Metzen and Chacron, 2014*; *Metzen et al., 2016*). Invariance scores for behavior were computed as described above for neural responses except that we used the behavioral PSTHs as responses within a time window of 1 s after stimulus chirp onset. Chirp rate was quantified as the average number of chirps emitted during a time window of 1 s after stimulus chirp onset divided by the time length of the window. Error bars were estimated using a sequential bootstrapping method (*Efron, 1979*) with block size 31 as done previously (*Metzen et al., 2016*).

### Statistics

Statistical significance was assessed through one-way analysis of variance (ANOVA) or Kruskal-Wallis test with the Bonferroni method of correcting for multiple comparisons at the $p=0.05$ level. Values are reported as mean ± SEM.

### Data availability

All data and codes are freely available from the Dryad Digital Repository: 10.5061/dryad.7pt59 (*Metzen and Chacron, 2017*).

## Acknowledgements

This research was supported by the Canadian Institutes of Health Research and the Canada Research Chairs (MJC).

## Additional information

### Funding

| Funder | Grant reference number | Author |
| --- | --- | --- |
| Canadian Institutes of Health Research | Operating grant | Maurice J Chacron |
| Canada Research Chairs | Canada Research Chair in Neural Information Coding | Maurice J Chacron |

The funders had no role in study design, data collection and interpretation, or the decision to submit the work for publication.

### Author contributions

MGM, Conceptualization, Resources, Data curation, Software, Formal analysis, Validation, Investigation, Visualization, Methodology, Writing—original draft, Writing—review and editing; MJC, Conceptualization, Resources, Supervision, Funding acquisition, Validation, Investigation, Writing—original draft, Project administration, Writing—review and editing

### Author ORCIDs

Michael G Metzen, http://orcid.org/0000-0002-2365-4192
Maurice J Chacron, http://orcid.org/0000-0002-3032-452X

### Ethics

Animal experimentation: All experimental procedures were approved by McGill University's animal care committee under protocol number 5285.

## Additional files

### Major datasets

The following dataset was generated:

| Author(s) | Year | Dataset title | Dataset URL | Database, license, and accessibility information |
| --- | --- | --- | --- | --- |
| Metzen MG, Chacron MJ | 2017 | Data from: Stimulus background influences phase invariant coding by correlated neural activity | http://dx.doi.org/10.5061/dryad.7pt59 | Available at Dryad Digital Repository under a CC0 Public Domain Dedication |

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
