## [Decision Letter]

Thank you for submitting your article "The effects of background on detection and phase invariant coding of transient natural communication signals by correlated neural activity" for consideration by *eLife*. Your article has been favorably evaluated by Eve Marder (Senior Editor) and three reviewers, one of whom, Ronald L Calabrese (Reviewer #3), is a member of our Board of Reviewing Editors. The following individual involved in review of your submission has agreed to reveal their identity: Bruce A Carlson (Reviewer #1).

The reviewers have discussed the reviews with one another and the Reviewing Editor has drafted this decision to help you prepare a revised submission.

Summary:

This manuscript is an advance on Metzen et al., 2016 in which the authors showed that correlations between the activities of peripheral afferents mediate a phase invariant representation of natural communication stimuli (chirps) that is refined across successive processing stages thereby leading to perception and behavior in the weakly electric fish *Apteronotus leptorhynchus*. In this advance, they explore how the phase invariance of neuronal responses and behavior to chirps is affected by background beat frequency determined by the difference in EOD frequency of two fish exhibiting steady EOD. The frequency contrast between the background beat and chirps decreases as the background beat frequency increases, but in contrast to previous studies that have focused on single neuron activity in afferents, in this study the authors show that it is correlations between the spiking activities of multiple afferents that determine behavioral responses. They show that the decreased signal detectability is due to the greater tendency of peripheral afferents to display phase locking to higher background beat frequencies, (a direct consequence of their high-pass frequency tuning characteristics). Such phase locking causes increased correlations among afferents during some portions of the background beat for which both afferents do not fire action potentials. This greater variability in correlation throughout the beat cycle then decreases signal detectability. The advance is well written and clearly illustrated, and it significantly extends the previous finding and gives new behavior relevance to afferent correlations, while explaining deterioration of behavioral responses with high frequency beat backgrounds.

Essential revisions:

There are some concerns in the detailed review of reviewer #1 (below) that must be addressed in revision. The most important are the technical points:

1) The measure of phase-locking (subsection “Electrophysiology”), which is based on a binary threshold decision.

2) The use of a sliding window for measuring correlations (subsection “Correlation between the spiking activities of electrosensory afferents”).

Moreover, there was a general concern among the reviewers about the major conclusion. The importance of correlation for invariant coding was demonstrated in the previous publication. The new result here is that the phase invariant coding breaks downs for high frequency beats. The fact that it breaks down at high frequency beats for both the afferent correlation coding and behavioral responses does strengthen the central claim of the first paper that correlated afferent firing mediates perception of the chirps. However, once behavior breaks down, we do not know whether it was because it relied on correlations or some other computation. Thus, the conclusion should be tempered by this caveat.

*Reviewer #1:*

The authors build on their recent finding of invariant coding of chirps with respect to chirp phase based on primary afferent correlations to address whether this coding of chirps is also invariant with respect to background frequency. They find that primary afferent correlations invariantly code for chirps only at relatively low background frequencies, and that this correlates with behavioral responses to chirps. This strengthens their previous conclusion that chirp perception is mediated by primary afferent correlations. This finding also meshes well with previous studies showing that chirps are more frequently produced on low-frequency, low-intensity backgrounds. This is a nice follow-up to the previous work, though I do have some concerns with the methods and conclusions.

1) The authors do a nice job of relating their findings to previous observations that chirps occur more frequently on lower baseline frequencies. However, I take issue with the authors' interpretation of cause and effect. They suggest that the inability to accurately encode chirps is the reason why the fish don't produce chirps at high baseline frequencies. It could be the exact opposite. If chirps don't occur on top of high-frequency beats, then why bother encoding them? I don't think the authors can conclude either way whether constraints of the sensory system are driving the behavior, or that behaviorally relevant signals are driving properties of the sensory system.

2) Related, do chirps vary quantitatively with baseline frequency? E.g. they may happen less frequently at high baseline frequencies, but do such chirps tend to involve higher frequency excursions? If so, then the use of a constant frequency change across baseline frequencies may not be a "natural stimulus." Whether or not chirps vary with baseline, it seems inaccurate to refer to all of these stimuli as "natural" if some of them tend not to occur naturally.

3) Subsection “Implications for other systems”: It is good to link the findings of this study to other sensory systems and taxonomic groups to make the case for general principles. However, it seems that these are fundamentally different kinds of invariance problems to solve. Detecting a chirp irrespective of background frequency or phase is different from detecting frequency irrespective of amplitude, or odor identity irrespective of concentration. The latter have nothing to do with "background." "Invariance" is being treated here as though it were a measurable physical thing, when really it is context-dependent and can refer to any kind of physical thing. The mechanisms for invariant coding are likewise likely to be context- and stimulus-dependent. This does not mean the findings do not have broad relevance, simply that they need to avoid implying that this is *the* solution to invariance.

4) Subsection “Implications for other systems”: Do *not* refer to animal species as higher or lower. This is simply not how evolution works. Further, if mammalian nervous systems can solve the problem, then so can fishy ones. It seems more likely that either: (i) these signals (chirp on high-frequency beat) are not behaviorally relevant, so there is no need to solve the problem, or (ii) that the problem of invariance being solved by these fish is a fundamentally different invariance problem from that solved by the other examples given.

5) I have concerns about the measure of phase-locking (subsection “Electrophysiology”). It is based on a binary threshold decision. Why is it done this way as opposed to a more standard, linear measure of vector strength (e.g. Goldberg and Brown 1969)? Your measure seems to nonlinearly exaggerate differences in the degree of phase-locking. Indeed, in Figure 2—figure supplement 1, you can clearly see phase-locking across all 3 stimuli (though different in degree), but the metric used gives you values = 0 at all frequencies <10 Hz in Figure 2—figure supplement 1. This is not a crucial metric for the conclusions reached, but it is somewhat perplexing why it is used.

6) I have concerns about the use of a sliding window for measuring correlations (subsection “Correlation between the spiking activities of electrosensory afferents”). If the window gets shorter as baseline frequency gets higher, then correlations might decrease artefactually due to both sampling fewer spikes and increasing the temporal precision needed for a correlation. A similar concern arises in the measurement of invariance across baseline frequencies, for which a window that varies with beat frequency is also used (subsection “Quantifying neural response invariance”). Regardless of this methodological issue, it's not clear how these variably sized windows are physiologically relevant. The integration time window of postsynaptic neurons that would detect these correlations should not change with beat frequency.

---

## [Author Response]

*Essential revisions:*

*There are some concerns in the detailed review of reviewer #1 (attached) that must be addressed in revision. The most important are the technical points:*

*1) The measure of phase-locking (subsection “Electrophysiology”), which is based on a binary threshold decision.*

We understand and agree with this concern and have added the rational why using this metric in the Methods and Results. In summary, we are interested in the phenomenon of phase locking in which spiking only occurs during some phases of the stimulus cycle and use a measure that reflects this definition. The full detail of our response can be found below.

*2) The use of a sliding window for measuring correlations (subsection “Correlation between the spiking activities of electrosensory afferents”).*

We have added more details on the rationale for using these sliding windows. To summarize, we now make clear that the values of the time windows used were directly measured from our experimental data (Figure 2—figure supplement 1). We also provide some citations to experimental work showing that the integration time windows of single neurons can change and also mention that decoding could alternatively be done by different subsets of downstream neurons. The full details of our response can be found below.

*Moreover, there was a general concern among the reviewers about the major conclusion. The importance of correlation for invariant coding was demonstrated in the previous publication. The new result here is that the phase invariant coding breaks downs for high frequency beats. The fact that it breaks down at high frequency beats for both the afferent correlation coding and behavioral responses does strengthen the central claim of the first paper that correlated afferent firing mediates perception of the chirps. However, once behavior breaks down, we do not know whether it was because it relied on correlations or some other computation. Thus, the conclusion should be tempered by this caveat.*

We understand this general concern and have revised our conclusions accordingly (subsection “Summary of results”).

*Reviewer #1:*

*[…] 1) The authors do a nice job of relating their findings to previous observations that chirps occur more frequently on lower baseline frequencies. However, I take issue with the authors' interpretation of cause and effect. They suggest that the inability to accurately encode chirps is the reason why the fish don't produce chirps at high baseline frequencies. It could be the exact opposite. If chirps don't occur on top of high-frequency beats, then why bother encoding them? I don't think the authors can conclude either way whether constraints of the sensory system are driving the behavior, or that behaviorally relevant signals are driving properties of the sensory system.*

We agree with the reviewer’s comment and have rephrased to mention that our results support the hypothesis that small chirps occur more rarely on top of beats with higher frequencies because they then cannot be encoded and perceived in a phase invariant manner. We also mention that it is also possible that small chirp stimuli on top of high frequency beats are not behaviorally relevant because they occur relatively infrequently in the first place.

*2) Related, do chirps vary quantitatively with baseline frequency? E.g. they may happen less frequently at high baseline frequencies, but do such chirps tend to involve higher frequency excursions? If so, then the use of a constant frequency change across baseline frequencies may not be a "natural stimulus." Whether or not chirps vary with baseline, it seems inaccurate to refer to all of these stimuli as "natural" if some of them tend not to occur naturally.*

In order to address this concern, we have performed additional analysis of the characteristics (i.e., duration and frequency excursion) of emitted chirps. The distribution of both measures obtained using different beat frequencies were not significantly different from one another (Figure 3—figure supplement 1). This is now mentioned in the Results.

*3) Subsection “Implications for other systems”: It is good to link the findings of this study to other sensory systems and taxonomic groups to make the case for general principles. However, it seems that these are fundamentally different kinds of invariance problems to solve. Detecting a chirp irrespective of background frequency or phase is different from detecting frequency irrespective of amplitude, or odor identity irrespective of concentration. The latter have nothing to do with "background." "Invariance" is being treated here as though it were a measurable physical thing, when really it is context-dependent and can refer to any kind of physical thing. The mechanisms for invariant coding are likewise likely to be context- and stimulus-dependent. This does not mean the findings do not have broad relevance, simply that they need to avoid implying that this is the solution to invariance.*

We understand the reviewer’s concern and have rewritten accordingly.

*4) Subsection “Implications for other systems”: Do not refer to animal species as higher or lower. This is simply not how evolution works. Further, if mammalian nervous systems can solve the problem, then so can fishy ones. It seems more likely that either: (i) these signals (chirp on high-frequency beat) are not behaviorally relevant, so there is no need to solve the problem, or (ii) that the problem of invariance being solved by these fish is a fundamentally different invariance problem from that solved by the other examples given.*

We have rewritten accordingly.

*5) I have concerns about the measure of phase-locking (subsection “Electrophysiology”). It is based on a binary threshold decision. Why is it done this way as opposed to a more standard, linear measure of vector strength (e.g. Goldberg and Brown 1969)? Your measure seems to nonlinearly exaggerate differences in the degree of phase-locking. Indeed, in Figure 2—figure supplement 1, you can clearly see phase-locking across all 3 stimuli (though different in degree), but the metric used gives you values = 0 at all frequencies <10 Hz in Figure 2—figure supplement 1. This is not a crucial metric for the conclusions reached, but it is somewhat perplexing why it is used.*

We understand the reviewer’s concern and now more clearly explain what we mean by phase locking. Specifically, we define as phase locking the tendency of neurons to fire action potentials reliably only during some range of phases of the sinusoidal stimulus cycle (see e.g. Keener et al., 1981; Trussell, 1999). As such, phase locking is a nonlinear phenomenon. Moreover, neurons that reliably display nonlinear rectification (i.e., are driven into cessation of firing) at some stimulus phases must then reliably fire action potentials at other stimulus phases must also display phase locking (see e.g. McGillivray et al. 2012). We thus used an index that quantifies the tendency of afferents to display rectification (i.e., complete cessation of firing) during some portions of the stimulus cycle. We note that measures like the vector strength will not capture such nonlinearities in general as they can be non-zero even when the sinusoidal stimulus elicits sinusoidal modulations in firing rate: a situation for which there is a linear relationship between stimulus and response. This explanation has been added to the Results and Methods.

*6) I have concerns about the use of a sliding window for measuring correlations (subsection “Correlation between the spiking activities of electrosensory afferents”). If the window gets shorter as baseline frequency gets higher, then correlations might decrease artefactually due to both sampling fewer spikes and increasing the temporal precision needed for a correlation. A similar concern arises in the measurement of invariance across baseline frequencies, for which a window that varies with beat frequency is also used (subsection “Quantifying neural response invariance”). Regardless of this methodological issue, it's not clear how these variably sized windows are physiologically relevant. The integration time window of postsynaptic neurons that would detect these correlations should not change with beat frequency.*

We understand the reviewer’s concerns. To address the first concern that correlations might decrease due to sampling of fewer spikes and increasing the temporal precision needed for a correlation, we now plot the un-normalized correlation coefficients in Figure 2 and note that correlations actually reach higher values for higher beat frequencies than for lower beat frequencies (Figure 2, compare dark and light purple curves). To address the second concern, we note that the lengths of the time windows were chosen based on our experimental data showing that the chirp response duration decreases with increasing beat frequency for both single neuron and correlated activity (Figure 2—figure supplement 1). This was done in order to ensure that the invariance scores computed only reflect responses to chirp rather than include the response to the beat, which can decrease invariance as shown in our previous publication (Metzen et al. (2016); Figure 2—figure supplement 1). Finally, to address the last concern about the physiological relevance of changing the analysis time window with beat frequency, we note that previous work has shown that the integration time window of sensory neurons can change based on stimulus attributes (Reid et al., 1992; Bair and Movshon, 2004; Prescott and De Koninck, 2005; Assisi et al., 2007; Butts et al., 2007; Ratté et al., 2013). It is also conceivable that information carried by afferent populations could be decoded by separate populations of hindbrain and midbrain neurons with different integration time windows. These arguments have been added to the Methods.